# An Evaluation of the ASTar Automated Antimicrobial Testing System for Gram-Negative Bacteria in Positive Blood Cultures

**DOI:** 10.3390/antibiotics14030249

**Published:** 2025-03-01

**Authors:** Hannah Coulson, Angelo Ivin, Kathryn M. Day, Danielle J. C. Fenwick, Emma C. L. Marrs, Patrick Mpwilu, John D. Perry

**Affiliations:** Microbiology Department, Freeman Hospital, Newcastle upon Tyne NE7 7DN, UK; hannah.coulson9@nhs.net (H.C.); angelo.ivin@nhs.net (A.I.); kathryn.day3@nhs.net (K.M.D.); danielle.fenwick@nhs.net (D.J.C.F.); e.marrs@nhs.net (E.C.L.M.); patrick.mpwilu@nhs.net (P.M.)

**Keywords:** blood culture, antimicrobial susceptibility testing, sepsis, bacteraemia

## Abstract

**Background**: Prompt administration of optimal antibiotic therapy is essential in the management of bacteraemia to reduce morbidity and mortality and to facilitate antibiotic stewardship. To identify the most effective therapy, rapid and accurate antimicrobial susceptibility testing (AST) is essential. ASTar is an automated AST system that delivers minimum inhibitory concentrations (MICs) for 23 antimicrobials and is designed for testing Gram-negative bacteria directly from positive blood cultures, with results available after 6 h. **Methods**: The ASTar system was evaluated with 64 positive blood cultures from patients with bacteraemia and 56 simulated blood cultures inoculated with a range of antibiotic-resistant isolates. The ASTar results for 12 antibiotics commonly used in our hospitals were compared with the results derived from three different methods of disc susceptibility testing and MICs determined by broth microdilution (BMD). **Results**: For 121 isolates of Gram-negative bacteria, ASTar showed an average essential agreement of 87.2% and an average categorical agreement of 94%, when compared with BMD. Very major errors (false susceptibility) and major errors (false resistance) were associated with 0.9% and 3.4% of results, respectively. The results were at least as accurate as those obtained from EUCAST disc susceptibility methods (both rapid and overnight methods). **Conclusions**: The ASTar system is an effective method for delivering accurate and rapid (6 h) AST results for Gram-negative bacteria by the direct testing of positive blood cultures.

## 1. Introduction

Bacteraemia is a severe systemic infection reported to affect over one million people in Europe each year, with a mortality rate estimated at 13% [1]. Gram-negative bacteria are frequently implicated. For example, in a study of 14,000 patients in intensive care in 75 countries, Gram-negative bacteria were isolated from 62% of patients with severe sepsis who had positive blood cultures [2]. Blood culture remains the cornerstone of diagnosis, but until antimicrobial susceptibility testing (AST) results are known, it is necessary to use broad-spectrum empirical therapy due to increasing levels of antibiotic resistance [3,4] and because delayed therapy with an effective agent may lead to an increase in mortality [5]. Once AST results are available, therapy may be safely de-escalated to more narrow-spectrum antibiotics [6]. Hence, rapid AST can facilitate the selection of effective early therapy and decisions that promote antibiotic stewardship [7,8,9,10].

Conventional AST relies on the availability of pure bacterial colonies to prepare an inoculum and tests that require a further 18–24 h of incubation to generate results [11,12]. This leads to unacceptable delays of 36–48 h to obtain AST results when every hour of delay can be critical, particularly in patients with septic shock [13]. Over the last decade, there has been a great deal of focus on technologies that are able to deliver rapid AST results directly from positive blood cultures—with a particular focus on Gram-negative pathogens. In some cases, genes encoding resistance mechanisms (e.g., for β-lactamases) are targeted by PCR [14,15], or the enzymes themselves may be targeted by phenotypic assays [16]. Such assays are useful to indicate antibiotics that should be avoided (due to the likelihood of resistance) but do not provide a reliable indication of whether a pathogen is susceptible to a particular antibiotic. Even when whole-genome sequencing (WGS) is available, in 2017 an expert group concluded that, for most bacterial species, there is currently insufficient evidence to support the use of WGS-inferred AST to guide clinical decision-making [17]. Although progress has been made, difficulties in the interpretation of the results remain [18], and the prediction of some susceptibility results using WGS (e.g., for amoxicillin–clavulanate against *Escherichia coli*) remains extremely challenging [19].

Given the challenges of predicting susceptibility using molecular methods, a range of phenotypic assays that measure bacterial growth have been developed for rapid AST directly from positive blood cultures. Examples of these include the EUCAST rapid AST (RAST) manual disc diffusion method that can generate AST results in as little as 4 h for common pathogens such as *Escherichia coli* and *Klebsiella pneumoniae* [20]. A range of automated systems can also provide rapid phenotypic AST results, with or without bacterial identification, and examples include the Accelerate Pheno system [21], FASTinov ultrarapid flow cytometry [22], QuickMIC [23], Alfred 60/AST [24], VITEK^®^ REVEAL^TM^ [25], and the dRAST system [26]. These systems, and alternative methods, have been the subject of a recent review [27].

The ASTar (ASTar; Q-linea AB, Uppsala, Sweden) is a relatively new fully automated rapid phenotypic test for the AST of Gram-negative bacteria in positive blood cultures [28,29,30]. Bacterial growth is monitored by the ASTar instrument via time-lapse microscopy, and growth (or inhibition) is interpreted via image analysis [28]. Species identification is not performed and is required for the interpretation of minimum inhibitory concentration (MIC) results, which are available for up to 23 antibiotics. The results are available after 6 h incubation. The aim of this study was to evaluate the ASTar with 120 blood cultures (64 patient samples and 56 simulated samples) and compare the results with reference results obtained by broth microdilution (BMD) [11] and our routine disc susceptibility testing methods [12,20].

## 2. Results

### 2.1. Species Recovered from Patient’s Blood Cultures and Reference MIC Results as Determined by BMD

Sixty-five Gram-negative isolates were recovered from 64 patients’ blood cultures. The most frequently identified species were *Escherichia coli* and *Klebsiella pneumoniae*. For most of these blood cultures (54/64), an acceptable score for species identification was obtained by performing matrix-assisted laser desorption/ionization–time-of-flight mass spectrometry (MALDI-TOF MS) directly on the blood culture extracts. The remainder were identified by testing colonies from subcultures after 4 h incubation. The species recovered are listed in Table 1, and their susceptibilities to 12 antibiotics, as determined by measurement of minimum inhibitory concentrations (MICs) using broth microdilution, are summarized in Table 2. Not all antibiotics were relevant to all species recovered. Four isolates of Gram-positive bacteria were also recovered and mixed with Gram-negative bacteria. These included *Enterococcus faecalis* (*n* = 2), *Enterococcus faecium* (*n* = 1), and *Staphylococcus epidermidis* (*n* = 1). None of these Gram-positive isolates was detected on the initial Gram stain of the positive blood culture or when MALDI-TOF MS was applied directly to the positive blood culture extracts. The 64 blood cultures included 36 aerobic (FA PLUS) bottles and 28 anaerobic (SN) bottles.

### 2.2. Species Recovered from Simulated Blood Cultures and Reference MIC Results as Determined by BMD

All Gram-negative species that were artificially inoculated into 56 simulated blood cultures were recovered as expected, with no additional unexpected isolates. Forty-seven of the isolates were successfully identified by the direct testing of blood cultures and the remainder from colonies obtained by 4 h subculture. The dominant species were *E. coli* and *K. pneumoniae*. There were equal numbers of aerobic (FA PLUS) bottles and anaerobic (SN) blood culture bottles. Table 3 summarizes the species that were recovered from the 56 simulated blood cultures, and their susceptibility to 12 antibiotics, as determined by a measurement of MICs using broth microdilution, is summarized in Table 4.

### 2.3. Performance of ASTar for 64 Routine Positive Blood Cultures from Patients

We first assessed the performance of ASTar for the susceptibility testing of Gram-negative bacteria in 64 patients’ blood cultures. Broth microdilution was used as the comparator. Sixty-five Gram-negative isolates were recovered, with one bottle containing a mixture of *E. coli* and *K. pneumoniae*. The performance data are summarized in Table 5. For 8/12 antibiotics, all relevant results were reported. Three results for ceftazidime against *Proteus mirabilis* were unreported by ASTar, as well as four results for ertapenem against *E. coli*. For colistin, the results were only reported against *E. coli* and *K. pneumoniae* and not for any other species, including, for example, *Pseudomonas aeruginosa* and *Acinetobacter baumannii*. No ASTar results were reported for trimethoprim–sulfamethoxazole against *A. baumannii*. Categorical agreement was very high between ASTar and BMD, with at least 94% for 10 out of the 12 antibiotics evaluated. The lowest categorical agreement was 79% for amoxicillin–clavulanate, and this was attributable to 11 major errors (ME), i.e., isolates falsely classified as resistant by ASTar. For ten of these isolates, the ASTar MIC was 16 mg/L, which is one dilution above the susceptibility breakpoint of 8 mg/L. For these 10 isolates (of which, 9 were *E. coli*), the MICs determined by BMD were typically 4–8 mg/L. Eleven other MEs were most likely attributable to the presence of a second species that was undetected by the initial Gram stain, including four species of Gram-positive bacteria. All MEs associated with aztreonam, ceftazidime–avibactam, ertapenem, meropenem, and trimethoprim–sulfamethoxazole were associated with polymicrobial cultures.

In total, MEs were associated with 4.1% of ASTar results. Very major errors (VMEs), indicating an incorrect ‘susceptible’ result, were very rare and were associated with only 2 out of 705 reported results (0.3%).

### 2.4. Performance of ASTar for 56 Simulated Blood Cultures

After testing samples from patients, we then assessed the performance of ASTar using 56 simulated blood cultures that had been inoculated with a range of Gram-negative species with known mechanisms of resistance to various antibiotics. Broth microdilution was again used as the comparator. The performance data are summarized in Table 6. As for the clinical blood cultures, the colistin results were only reported against *E. coli* and *K. pneumoniae* and no ASTar results were reported for trimethoprim–sulfamethoxazole against *A. baumannii*. Categorical agreement was high between the ASTar and BMD: at least 92% for 11 out of the 12 antibiotics. The categorical agreement was lowest (75%) for meropenem, in large part due to eight major errors indicating a false indication of meropenem resistance. All of these eight isolates possessed carbapenemase enzymes capable of hydrolyzing meropenem. The ASTar MICs for these eight isolates ranged from 16 to 64 mg/L (resistant), whereas BMD MICs ranged from 4 to 8 mg/L (SIE). There were nine VMEs out of 553 reported results (1.6%).

### 2.5. Performance of ASTar in Comparison with Overnight Disc Susceptibility Testing

Two methods of overnight disc susceptibility testing were evaluated in this study, including the standard EUCAST disc method performed from colonies recovered from blood cultures and an in-house method that was applied to the direct testing of positive blood cultures. The performances of both of these methods were compared to that of ASTar in Table 7 and Table 8. All test results were assessed against BMD, and for convenience, the tables summarize the data for all blood cultures (i.e., both patient samples and simulated samples combined). Colistin is not included in these tables, as no interpretive breakpoints are available for disc testing with colistin.

ASTar showed a higher level of categorical agreement when compared with direct disc susceptibility testing for 7 out of 11 antibiotics. VMEs were associated with 9 ASTar results out of 1175 total results reported for these 11 antibiotics (0.8%). This is compared with 2% for the EUCAST disc method and 1.7% of the reported results for the direct disc method.

### 2.6. Performance of ASTar in Comparison with RAST

The EUCAST RAST method is a disc susceptibility test that allows for the determination of rapid AST results in as little as 4 h. RAST breakpoints are only published for a limited range of antibiotics—including 8 of the 12 antibiotics utilized in this study. A subset of 101 blood cultures were subjected to RAST, including all simulated blood cultures (*n* = 56) and 45 blood cultures from patients. The subset included Enterobacterales (*n* = 85), *P. aeruginosa* (*n* = 9), and *A. baumannii* complex (*n* = 7). Table 9 compares the categorical agreement with BMD for both ASTar and RAST and indicates a very similar level of performance for both methods. The occurrence of VMEs for both methods is summarized in Table 10 and reveals a lower proportion of VMEs for ASTar (1.1% of reported results) compared with RAST (2.9% of reported results). Most notably, there were fewer VMEs for piperacillin–tazobactam.

Table 11 shows the time taken to generate a result for ASTar and RAST. ASTar only generates results after 6 h of incubation, but this allowed for the reporting of an average of 99.5% of the relevant results for the eight antibiotics. An average of 81.6% of the relevant results were available by RAST after 4 h of incubation, rising to 92.7% after 6 h. RAST was not incubated beyond 6 h. RAST delivered the lowest percentage (82.4%) of relevant results for amoxicillin–clavulanate due to a large number of indeterminate results falling into the area of technical uncertainty (ATU).

## 3. Discussion

There are three previous published reports on the performance of the ASTar instrument. In a large study, published in 2023 by Göransson et al., the performance of the ASTar system was assessed with 412 simulated blood cultures and 74 clinical blood cultures [28]. They reported that the ASTar system demonstrated an overall categorical agreement of 97.6% and an essential agreement of 95.8% compared to the BMD reference method. Also in 2023, Esse et al. reported the findings of a smaller study with a total of 78 blood cultures, including 51 clinical samples and 27 simulated blood cultures containing isolates with resistance mechanisms. The categorical and essential agreements were 95.6% and 90.7%, respectively, when compared with the BMD [29]. Finally, in 2024, Banchini et al. analyzed a total of 43 clinical blood cultures using the ASTar and compared the results for 15 antibiotics with those obtained by a commercial microdilution assay (MicroScan Walkaway). The categorical and essential agreements were 96.1% and 98%, respectively [30]. Our findings, with an overall categorical agreement of 94% and an essential agreement of 87%, are closest to those reported by Esse et al. [29].

It is worth noting that there are some differences in the methods for data analysis between different studies. Just as one example, in at least two of these studies [28,29], the samples that were initially unrecognized as polymicrobial cultures were excluded from the final analysis. We elected to include such samples, as the ASTar results would be utilized routinely after 6 h ‘in real-life’. However, the presence of a mixed culture would likely not be known until at least 12 h later. This is one reason for the slightly lower figures for essential and categorical agreement in our study. For example, in this study, five samples that turned out to be polymicrobial were associated with 42% of all categorical errors, including 38% of all major errors, when testing clinical blood cultures. Polymicrobial cultures did not account for very major errors and were not found in simulated blood cultures.

This problem could potentially be reduced by a more thorough and careful reading of Gram-stained smears to exclude the presence of Gram-positive bacteria. The complete antibiogram should also be fully considered when assessing the validity of the results. For example, a polymicrobial blood culture containing *E. cloacae* complex and *E. faecalis* showed resistance to meropenem (attributable to *E. faecalis*) but susceptibility to piperacillin–tazobactam, when tested by ASTar. This represents a highly unusual antibiogram for a pure isolate of *E. cloacae* and should raise suspicions of the possibility of a polymicrobial culture.

In the study by Esse et al., it is interesting to note that the highest number of major errors were associated with amoxicillin–clavulanate (major error rate: 23.1%) and that these errors were almost exclusively associated with *E. coli* [29]. These findings are consistent with our own observations when testing clinical blood cultures. We reported a total of 11 major errors for amoxicillin–clavulanate-susceptible isolates when testing patient’s blood cultures using ASTar (*E. coli (n* = 9), *K. pneumoniae* (*n* = 1), and *P. mirabilis* (*n* = 1)). Notably, all 11 of these isolates were highly resistant to ampicillin (MIC > 64 mg/L) when tested by ASTar, inferring β-lactamase production, which suggests that the clavulanate component of amoxicillin–clavulanate may be critically relevant to these errors. The vast majority (52/56) of the isolates used for the preparation of simulated blood cultures were highly resistant to amoxicillin–clavulanate (MIC > 32 mg/L), so the likelihood of major errors was much lower.

From 64 patients’ blood cultures, we did not encounter any Gram-negative isolates that were resistant to meropenem, and the categorical agreement between ASTar and BMD was high (97%). In contrast, when testing 56 simulated blood cultures, there were 8 major errors and 2 very major errors. All of these 10 errors were associated with strains producing a variety of carbapenemases. The two Enterobacterales showing very major errors had ASTar MICs of 4 mg/L (SIE) versus MICs of 16 mg/L (R) when tested by BMD. Major errors were associated with *A. baumannii* (*n* = 3), *P. aeruginosa* (*n* = 1), and various Enterobacterales (*n* = 4). ASTar MICs for these isolates were 16–64 mg/L (R) versus MICs of 4–8 mg/L (SIE) when tested by BMD. The reasons for the relatively poor correlation for meropenem when testing carbapenemase producers are unclear. Esse et al. did not report any specific data for carbapenemase producers but reported a categorical agreement of 90.7% for meropenem, which was relatively low when compared to other antibiotics [29]. It is not known whether any carbapenemase producers were included in the study by Göransson et al. [28], and all isolates in the study by Banchini et al. were susceptible to carbapenems [30]. Further studies with carbapenemase-producing isolates would be of value.

In our laboratory, we have routinely performed the EUCAST RAST method for blood cultures containing Gram-negative bacteria for the past three years. For eight of the antibiotics evaluated in this study, we were able to generate susceptibility data due to the availability of published EUCAST breakpoints. The average categorical agreement for RAST and ASTar was 93.9% and 92.8%, respectively, for these eight antibiotics. However, there was a higher proportion of ATU results for the RAST method, which meant that RAST delivered only 92.7% of the relevant results, compared with 99.5% for ASTar. Furthermore, if the EUCAST guidelines are strictly followed, interpretation of susceptibility is not available for Enterobacterales other than *E. coli* and *K. pneumoniae* [20]. In this study, and in routine practice, we apply *K. pneumoniae* breakpoints for all Enterobacterales other than *E. coli*, as we have shown this works well. But, this is a deviation from EUCAST guidelines. A major advantage of ASTar, when compared with RAST, is that the results are available for a wide range of antibiotics (up to 23) for a wider range of species. Furthermore, the instrument is extremely easy to use, with only a few minutes of staff time required for the inoculation of samples and the retrieval of results.

A weakness of this study is that we only elected to evaluate the 12 antibiotics that were of most relevance for the treatment of bloodstream infections in our hospital, whereas the ASTar instrument is able to deliver MIC results for up to 23 antimicrobials. Also, all of the clinical blood cultures were derived from a single centre. The strengths of this study are the inclusion of the ‘gold standard’ BMD method for the validation of the results, and the inclusion of both routine blood cultures and a challenging collection of isolates with a wide range of resistance mechanisms.

## 4. Materials and Methods

### 4.1. Patient Samples

Sixty-four consecutive positive blood cultures that showed an apparent pure culture of Gram-negative bacilli on Gram stain were included in the study. Only one bottle was selected per patient. After the bottles were removed from the Bact-Alert 3D instrument (bioMérieux, Durham, NC, USA) and the Gram stain was confirmed, the bottles were mixed by hand, and 5 mL were removed into a sterile bottle to use in the study. All study samples were anonymized so that no patient information was available, and the study results could not affect patient management.

### 4.2. Bacterial Identification

A 1 mL aliquot was transferred to a centrifuge tube, and 200 µL of saponin was added. This was then centrifuged at 3000 rpm for 1 min in a MiniSpin^®^ centrifuge (Eppendorf, Stevenage, UK). The supernatant was removed with a pipette and the deposit was re-suspended by vortexing after the addition of 2 mL of sterile deionized water (SDW). Centrifugation was repeated, and the supernatant was removed. The deposit was subjected to matrix-assisted laser desorption/ionization–time-of-flight mass spectrometry (MALDI-TOF MS), in accordance with the manufacturer’s instructions (Bruker, Coventry, UK) to achieve species identification. A 100 µL aliquot of the positive blood culture was inoculated onto Columbia blood agar (CBA; Thermofisher, Basingstoke, UK), and the inoculum was spread to obtain isolated colonies. This was initially incubated for 4 h at 37 °C in air and then overnight. For the ~15% of samples where direct identification was unsuccessful, MALDI-TOF MS was then re-attempted from colonies growing on CBA after 4 h.

### 4.3. Preparation of Simulated Blood Cultures

A collection of 56 Gram-negative isolates comprising 44 Enterobacterales (including *C. freundii* (*n* = 2), *E. cloacae* complex (*n* = 7), *E. coli* (*n* = 12), *K. pneumoniae* (*n* = 23)), *Pseudomonas aeruginosa* (*n* = 6), and *Acinetobacter baumannii* (*n* = 6) were retrieved from our archive collection and subcultured onto CBA for the preparation of simulated blood cultures. The strains selected for the artificial inoculation of blood cultures had various resistance mechanisms, including extended-spectrum β-lactamases, AmpC β-lactamases, carbapenemases, and mechanisms of colistin resistance (including *mcr* genes). A detailed list is shown in Appendix A. The plates were incubated overnight at 37 °C in air. Each isolate was inoculated into an anonymized negative blood culture that had been incubated for 5 days on the Bact-Alert 3D instrument (bioMérieux, Durham, NC, USA). Equal numbers of aerobic (FA PLUS) and anaerobic (SN) bottles were selected for inoculation.

For each isolate, colonies were suspended in 2 mL SDW, and the turbidity was adjusted to 0.5 McFarland units using a Densimat (bioMérieux, Basingstoke, UK) to obtain an inoculum of approximately 1–1.5 × 10^8^ CFU/mL. Two serial 1/200 dilutions were then performed (by adding 100 µL to 19.9 mL SDW), and a 25 µL aliquot of the resulting suspension was cultured on CBA to obtain a colony count (target: approx. 60 colonies). A further 1/10 dilution was performed (by adding 1 mL of suspension to 9 mL of SDW) to produce a suspension containing approximately 250 CFU/mL. Using a sterile syringe, 1 mL of this suspension was added to a negative blood culture that was then loaded back onto the Bact-Alert 3D instrument until it signalled as positive. The positive bottles were then processed as described for the patient samples.

### 4.4. ASTar Testing

A 1 mL sample of each positive blood culture was loaded onto the fully automated ASTar instrument in exact accordance with the manufacturer’s instructions. Each sample required 2–3 min to load. The species identification was added to the ASTar database as soon as it was available in order to allow for the MIC results to be interpreted by the instrument. The MIC results, which were available after 6 h incubation, were documented on an Excel spreadsheet.

### 4.5. EUCAST RAST Disc Susceptibility Testing

For the study samples that signalled as positive before 11 am, RAST was set up in exact accordance with EUCAST methodology [20]. Briefly, 125 µL of positive blood culture was removed and inoculated onto each of two Mueller–Hinton agar plates (Thermofisher, Basingstoke, UK). The inoculum was spread with a swab using a rotary spreader, and the discs containing the 8 antibiotics listed in Table 9 were added to the surface of the agar. After 4 h and 6 h of incubation at 37 °C in air, inhibition zone diameters were measured with a ruler, documented on an Excel spreadsheet, and interpreted according to EUCAST breakpoints for short incubation (version 7.2; 23 September 2024) [20]. In a deviation from the EUCAST method, we applied published breakpoints for *K. pneumoniae* for any species of Enterobacterales that had no published breakpoints.

### 4.6. Direct Disc Susceptibility Testing

An in-house method for direct susceptibility testing is routinely used in our laboratory and was included for comparison. One drop of positive blood culture (approx. 30 µL) was added to 5 mL of SDW. A swab was then immersed in this suspension and, after the removal of excess fluid, used to spread each of the two Mueller–Hinton agar plates using a rotary spreader. The discs containing the 11 antibiotics listed in Table 7 were then added to the surface of the agar. After 18–24 h of incubation at 37 °C in air, inhibition zone diameters were measured with a ruler, documented on an Excel spreadsheet, and interpreted according to standard EUCAST breakpoints for 18–24 h incubation (version 14; 1 January 2024). [12].

### 4.7. EUCAST Disc Susceptibility Testing

This was performed exactly as recommended by EUCAST [12]. Colonies on the CBA plates that were incubated overnight (see Section 4.2) were used to prepare a 0.5 McFarland inoculum, as described in Section 4.2 for inoculation of two Mueller–Hinton agar plates using a rotary spreader. The discs containing the 11 antibiotics listed in Table 7 were then added to the surface of the agar. After 18–24 h of incubation at 37 °C in air, inhibition zone diameters were measured with a ruler, documented on an Excel spreadsheet, and interpreted according to standard EUCAST breakpoints for 18–24 h incubation (version 14; 1 January 2024). [12].

### 4.8. Broth Microdilution Testing (BMD)

Colonies on the CBA plates that were incubated overnight (see Section 4.2) were stored at 4 °C and subcultured once only onto a fresh CBA plate on the day before BMD was performed. MICs were determined for the 12 antibiotics listed in Table 6 against all Gram-negative isolates recovered from the 64 patients’ blood cultures and the 56 simulated blood cultures. Antibiotic powders and β-lactamase inhibitors were purchased from Discovery Fine Chemicals, Wimborne, UK, and adjusted for potency when weighing. MIC tests were performed in accordance with the ISO 20776-1 method [11]. Briefly, this involved using Mueller–Hinton broth (Thermofisher, Basingstoke, UK) incorporating seven different double dilutions of each antibiotic and a final inoculum of 5 × 10^5^ CFU/mL (acceptable range 2–8 × 10^5^ CFU/mL). The β-lactamase inhibitors avibactam and tazobactam were incorporated into the relevant wells at a final concentration of 4 mg/L and potassium clavulanate at a final concentration of 2 mg/L, after adjusting for potency. The final volume in each microtitre well was 100 µL. MICs were read using a Synergy HT spectrophotometric plate reader (Agilent Technologies, Stockport, UK) after 18 h of incubation at 37 °C in air, and the MIC results were interpreted in accordance with EUCAST breakpoints for 18–24 h incubation (version 14; 1 January 2024) [12]. Any isolates that were susceptible to a particular antibiotic when tested by the ASTar instrument but resistant when tested by BMD, were re-tested by BMD to confirm the reproducibility of the reference result.

### 4.9. Quality Control

No test results were interpreted unless the results for the quality control strains were within specification. The following strains were utilized for quality control of the EUCAST disc susceptibility testing method [12]: *E. coli* (NCTC 12241), *E. coli* (NCTC 11954), *K. pneumoniae* (NCTC 13368), and *P. aeruginosa* (NCTC 12903). These strains were also included with every batch of BMD tests, along with an additional colistin-resistant strain: *E. coli* NCTC 13846. The results for the inhibition zone diameters and MICs were compared with the ranges permitted by EUCAST [31]. *E. coli* (NCTC 12241) and *P. aeruginosa* (NCTC 12903) were used for the quality control of RAST, as recommended [20].

For ASTar, the following control strains were tested with each batch of kits, in accordance with the manufacturer’s instructions: *E. coli* NCTC 12241, *P. aeruginosa* NCTC 12903, *K. pneumoniae* NCTC 13368, *K. pneumoniae* ATCC BAA 2814, and *Streptococcus pneumoniae* NCTC 12977.

### 4.10. Data Analysis

Each of the four AST methods was compared to the results obtained by BMD. The performance of each test was assessed using the principles recommended by the United States Food and Drug Administration [32] by calculation of essential agreement, categorical agreement, and the rate of major errors and very major errors. These were defined as follows:

**Essential agreement (EA)**: the percentage of isolates for which the MIC result (where available) was within one double dilution of the BMD result.

**Categorical agreement (CA)**: the percentage of isolates for which the category of result (i.e., susceptible (S) or resistant (R) or susceptible-increased exposure (SIE)) was identical to the category of result found using BMD.

**Very major error (VME)**: a VME occurred when the test method generated a susceptible result (S or SIE), and the result was R when tested by BMD. The rate of VME (%) for any particular antibiotic was calculated as follows:(Total number of very major errors/Total number of resistant isolates using BMD) × 100.

**Major error (ME)**: an ME occurred when the test method generated an R result, and the result was susceptible (S or SIE) when tested by BMD. The rate of ME (%) for any particular antibiotic was calculated as follows:(Total number of major errors/Total number of susceptible isolates using BMD) × 100.

**Minor error:** Where the test method generated an SIE result and the BMD result was S (or vice versa), for the purposes of this study, this was classified as a minor error. The rate of minor errors for any particular antibiotic was calculated as follows:(Total number of minor errors/Total number of isolates tested using BMD) × 100.

## 5. Conclusions

The ASTar system is an effective method for delivering accurate and rapid (6 h) AST results for Gram-negative bacteria by the direct testing of positive blood cultures. The instrument is extremely easy to use with only a few minutes of staff time required to run each sample. We report a high categorical agreement of 94% when compared with BMD and superior results to routinely performed disc diffusion methods. Reports of false susceptibility (very major errors) were associated with <1% of test results.

## Figures and Tables

**Table 1 antibiotics-14-00249-t001:** Species recovered from patients’ blood cultures.

	*n*
*Acinetobacter* species	2
*Citrobacter koseri*	1
*Enterobacter cloacae* complex	2
*Escherichia coli*	33
*Klebsiella aerogenes*	2
*Klebsiella pneumoniae*	15
*Pseudomonas aeruginosa*	4
*Proteus mirabilis*	3
*Serratia marcescens* complex	3
Total:	65

**Table 2 antibiotics-14-00249-t002:** Susceptibility of Gram-negative isolates from patient’s blood cultures (*n* = 65).

	CIP	GEN	CAZ	TAZ	MER	SXT	CAVI	AMC	ERT	AZT	CXM	COL
Total	65	61	63	63	65	61	63	52	59	63	51	65
R	13	10	8	6	0	23	0	18	2	12	16	6
S	46	51	47	53	63	38	63	34	57	46	0	59
SIE	5	0	8	4	2	0	0	0	0	5	35	0
ATU	1	0	0	0	0	0	0	0	0	0	0	0

Abbreviations: R: resistant; S: susceptible; SIE: susceptible-increased exposure; ATU: area of technical uncertainty; CIP: ciprofloxacin; GEN: gentamicin; CAZ: ceftazidime; TAZ: piperacillin–tazobactam; MER: meropenem; SXT: trimethoprim–sulfamethoxazole; CAVI: ceftazidime––avibactam; AMC: amoxicillin–clavulanate; ERT: ertapenem; AZT: aztreonam; CXM; cefuroxime; COL: colistin.

**Table 3 antibiotics-14-00249-t003:** Species recovered from simulated blood cultures.

	*n*
*Acinetobacter* species	6
*Citrobacter freundii*	2
*Enterobacter cloacae* complex	7
*Escherichia coli*	12
*Klebsiella pneumoniae*	23
*Pseudomonas aeruginosa*	6
Total:	56

**Table 4 antibiotics-14-00249-t004:** Susceptibility of isolates (*n* = 56) from simulated blood cultures.

	CIP	GEN	CAZ	TAZ	MER	SXT	CAVI	AMC	ERT	AZT	CXM	COL
Total	56	50	50	50	56	50	50	35	44	50	35	56
R	41	24	45	47	22	40	24	34	33	41	35	9
S	12	26	4	3	21	10	26	1	11	5	0	47
SIE	2	0	1	0	13	0	0	0	0	4	0	0
ATU	1	0	0	0	0	0	0	0	0	0	0	0

Abbreviations: R: resistant; S: susceptible; SIE: susceptible-increased exposure; ATU: area of technical uncertainty; CIP: ciprofloxacin; GEN: gentamicin; CAZ: ceftazidime; TAZ: piperacillin–tazobactam; MER: meropenem; SXT: trimethoprim–sulfamethoxazole; CAVI: ceftazidime–avibactam; AMC: amoxicillin–clavulanate; ERT: ertapenem; AZT: aztreonam; CXM; cefuroxime; COL: colistin.

**Table 5 antibiotics-14-00249-t005:** Performance of ASTar for susceptibility testing of 65 Gram-negative isolates from patients’ blood cultures.

	CIP	GEN	CAZ	TAZ	MER	SXT	CAVI	AMC	ERT	AZT	CXM	COL
Relevant results reported (%)	100	100	95.2	100	100	97	100	100	93.2	100	100	87.3
Minor errors	3	0	3	0	0	0	0	0	0	3	0	0
Major errors (ME)	0	0	5	4	2	1	2	11	1	1	2	0
Very major errors (VME)	0	1	0	0	0	0	0	0	0	1	0	0
Essential agreement (%)	92.3	100	80	84.1	96.9	93.2	90.5	78.8	87.3	90.5	78.4	97.9
Categorical agreement (%)	94	98	86.7	94	97	98	97	79	98.2	92	96	100
Minor error rate (%)	4.6	0	5.0	0	0	0	0	0	0	4.8	0	0
ME rate (%)	0	0	9.6	7.0	3.1	2.8	3.2	32.4	1.8	2.0	5.7	0
VME rate (%)	0	10	0	0	0	0	0	0	0	8.3	0	0

Abbreviations: CIP: ciprofloxacin; GEN: gentamicin; CAZ: ceftazidime; TAZ: piperacillin–tazobactam; MER: meropenem; SXT: trimethoprim–sulfamethoxazole; CAVI: ceftazidime–avibactam; AMC: amoxicillin–clavulanate; ERT: ertapenem; AZT: aztreonam; CXM; cefuroxime; COL: colistin.

**Table 6 antibiotics-14-00249-t006:** Performance of ASTar for susceptibility testing of 56 Gram-negative isolates from simulated blood cultures.

	CIP	GEN	CAZ	TAZ	MER	SXT	CAVI	AMC	ERT	AZT	CXM	COL
Relevant results reported (%)	98.2	100	100	100	100	88	98	100	100	100	100	62.5
Minor errors	3	0	1	0	4	0	0	0	0	0	0	0
Major errors (ME)	0	3	1	0	8	0	0	0	1	1	0	0
Very major errors (VME)	0	1	0	0	2	3	0	1	0	0	0	2
Essential agreement (%)	90.9	88	94	82	75	84.1	77.6	97.1	88.6	90	97.1	88.6
Categorical agreement (%)	93	92	96	100	75	93	100	97	97.7	98	100	94.3
Minor error rate (%)	5.5	0	2	0	7.1	0	0	0	0	0	0	0
ME rate (%)	0	11.5	20	0	23.5	0	0	0	9.1	11.1	0	0
VME rate (%)	0	4	0	0	9	8	0	3	0	0	0	18

Abbreviations: CIP: ciprofloxacin; GEN: gentamicin; CAZ: ceftazidime; TAZ: piperacillin–tazobactam; MER: meropenem; SXT: trimethoprim–sulfamethoxazole; CAVI: ceftazidime–avibactam; AMC: amoxicillin–clavulanate; ERT: ertapenem; AZT: aztreonam; CXM; cefuroxime; COL: colistin.

**Table 7 antibiotics-14-00249-t007:** Categorical agreement with BMD (%) for three different methods of AST for Gram-negative blood culture isolates (*n* = 121).

	CIP	GEN	CAZ	TAZ	MER	SXT	CAVI	AMC	ERT	AZT	CXM
ASTar	93.3	95.5	91	96.5	86.8	96.1	98.2	86.2	98	94.7	97.7
EUCAST disc method	95.7	95.5	92.9	92.7	80.2	95.4	100	94.5	93.2	93.8	98.8
Direct disc method	98.3	94.5	95	93.7	85	94.2	100	92.3	94.2	93.8	96.5

Abbreviations: CIP: ciprofloxacin; GEN: gentamicin; CAZ: ceftazidime; TAZ: piperacillin–tazobactam; MER: meropenem; SXT: trimethoprim–sulfamethoxazole; CAVI: ceftazidime–avibactam; AMC: amoxicillin–clavulanate; ERT: ertapenem; AZT: aztreonam; CXM; cefuroxime.

**Table 8 antibiotics-14-00249-t008:** Number of very major errors for three different methods of AST for Gram-negative blood culture isolates (*n* = 121).

	CIP	GEN	CAZ	TAZ	MER	SXT	CAVI	AMC	ERT	AZT	CXM	Total
ASTar	0	2	0	0	2	3	0	1	0	1	0	9
EUCAST disc method	2	3	1	4	3	3	0	0	2	4	1	23
Direct disc method	0	4	0	3	2	5	0	0	2	1	3	20

Abbreviations: CIP: ciprofloxacin; GEN: gentamicin; CAZ: ceftazidime; TAZ: piperacillin–tazobactam; MER: meropenem; SXT: trimethoprim–sulfamethoxazole; CAVI: ceftazidime–avibactam; AMC: amoxicillin–clavulanate; ERT: ertapenem; AZT: aztreonam; CXM; cefuroxime.

**Table 9 antibiotics-14-00249-t009:** Categorical agreement with BMD (%) for ASTar and RAST for Gram-negative blood culture isolates (*n* = 101).

	CIP	GEN	CAZ	TAZ	MER	SXT	CAVI	AMC
ASTar	93	94.6	93.5	93.6	85.2	95.3	98.9	88.7
RAST	95.7	91.9	100	91.6	85.4	92.7	98.9	94.6

Abbreviations: CIP: ciprofloxacin; GEN: gentamicin; CAZ: ceftazidime; TAZ: piperacillin–tazobactam; MER: meropenem; SXT: trimethoprim–sulfamethoxazole; CAVI: ceftazidime–avibactam; AMC: amoxicillin–clavulanate.

**Table 10 antibiotics-14-00249-t010:** Number of very major errors for ASTar and RAST for Gram-negative blood culture isolates (*n* = 101).

	CIP	GEN	CAZ	TAZ	MER	SXT	CAVI	AMC	Total
ASTar	0	2	0	0	2	3	0	1	8
RAST	1	4	0	7	2	5	0	1	20

Abbreviations: CIP: ciprofloxacin; GEN: gentamicin; CAZ: ceftazidime; TAZ: piperacillin–tazobactam; MER: meropenem; SXT: trimethoprim–sulfamethoxazole; CAVI: ceftazidime–avibactam; AMC: amoxicillin–clavulanate.

**Table 11 antibiotics-14-00249-t011:** Percentage of relevant results available by RAST after 4 h and 6 h of incubation compared to ASTar for a subset of 101 blood culture isolates.

	CIP	GEN	CAZ	TAZ	MER	SXT	CAVI	AMC
R-AST (4 h)	84	87	84	77.7	70.3	87	86.2	76.5
R-AST (6 h)	93.6	94.6	97.9	89.4	90.1	94.6	98.9	82.4
ASTar (6 h)	99	100	97.9	100	100	100	98.9	100

Abbreviations: CIP: ciprofloxacin; GEN: gentamicin; CAZ: ceftazidime; TAZ: piperacillin–tazobactam; MER: meropenem; SXT: trimethoprim–sulfamethoxazole; CAVI: ceftazidime–avibactam; AMC: amoxicillin–clavulanate.

## Data Availability

The raw data supporting the conclusions of this article will be made available by the authors upon request.

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
