# Peer review of "An Evaluation of the ASTar Automated Antimicrobial Testing System for Gram-Negative Bacteria in Positive Blood Cultures"

_antibiotics, 2025, doi:10.3390/antibiotics14030249_

Round 1
Reviewer 1 Report
Comments and Suggestions for Authors
- Summary:
The article titled “An Evaluation of the ASTar Automated Antimicrobial Testing System for Gram-negative Bacteria in Positive Blood Cultures” assesses the performance of the ASTar system in delivering rapid and accurate antimicrobial susceptibility testing results directly from blood cultures against a range of traditional AST methods. The study involved evaluating both clinical and simulated positive blood cultures for Gram-negative bacteria. Although the study highlights the reliable agreement of the ASTar system, which in itself is not entirely novel, it has the potential to more deeply explore the incidences of major and very major errors in relation to specific antimicrobial drugs and resistance mechanisms.
- Comments:
- Introduction:
- Line 56 – 57: avoid repetition of “rapid” in this sentence.
- Methods/Results:
- Simulated blood cultures (and the related Table 3 in results): In the methods you should make it clear how many samples were inoculated with which bacterial species (and how this relates to the resistance mechanism – table 4 indicates not all simulated samples were phenotypically resistant by BMD). Is recovery of organisms as declared in Table 3 different from the inoculated organism (i.e. do not all simulated blood samples positively recover / identify)?
- Discussion:
- Critical comparison with literature: the authors have done well to highlight other studies that have performed similar studies using the ASTar. However, without reading these other articles in detail the reader cannot identify the similarities/differences between this study and those others, did they all explore the same antimicrobials, did others show similar findings for antimicrobials of interest (i.e. Amoxi-clav; meropenem).
- Polymicrobial cultures were highlighted as the most probable cause for VME’s. Discussion should explore this further by providing more detailed insights into how ASTar differentiates mixed cultures and potential limitations arising from these cases, and/or how this can be addressed in future experiments.
- While the inclusion of resistant strains is noted, more quantitative data about how ASTar handled complex resistance mechanisms (like β-lactamases) would enhance the robustness. I.e do specific resistance mechanisms more commonly result in ME/VME’s….this is briefly touched on in the results for meropenem and should be discussed in more detail.
- Can the authors posit reasons for the poor performance with Amoxi-clav (ME’s in patient samples), meropenem, gentamicin, ceftazidime, and aztreonam (ME’s in simulated samples), consider ways that these may be overcome, or future studies to address these?
Author Response
Responses to reviewer 1
The authors are grateful for your diligent and helpful review. Please see our responses below.
Summary:
The article titled “An Evaluation of the ASTar Automated Antimicrobial Testing System for Gram-negative Bacteria in Positive Blood Cultures” assesses the performance of the ASTar system in delivering rapid and accurate antimicrobial susceptibility testing results directly from blood cultures against a range of traditional AST methods. The study involved evaluating both clinical and simulated positive blood cultures for Gram-negative bacteria. Although the study highlights the reliable agreement of the ASTar system, which in itself is not entirely novel, it has the potential to more deeply explore the incidences of major and very major errors in relation to specific antimicrobial drugs and resistance mechanisms.
Comments:
Introduction:
Line 56 – 57: avoid repetition of “rapid” in this sentence.
Author response: This has been corrected
Methods/Results:
Simulated blood cultures (and the related Table 3 in results): In the methods you should make it clear how many samples were inoculated with which bacterial species (and how this relates to the resistance mechanism – table 4 indicates not all simulated samples were phenotypically resistant by BMD).
Author response: We have added information in line 328 to show how many of each species were inoculated into simulated blood culture bottles. Appendix A all provides full details of which known resistance mechanisms was present in each individual species. The reviewer has stated that “table 4 indicates not all simulated samples were phenotypically resistant by BMD”. It is clearly the case (from Table 4) that every isolate recovered from simulated blood cultures was resistant to at least one antibiotic. For example, all isolates from simulated blood cultures were resistant to cefuroxime and all-but-one were resistant to amoxicillin-clavulanate. There should be no expectation that all isolates would be resistant to all antibiotics.
Is recovery of organisms as declared in Table 3 different from the inoculated organism (i.e. do not all simulated blood samples positively recover / identify)?
Author response: We recovered everything that we inoculated and nothing more. We see the benefit of making this clearer and have added the following sentence to line 104: “All species that were artificially inoculated into simulated blood cultures were recovered as expected, with no additional unexpected isolates”.
Discussion:
Critical comparison with literature: the authors have done well to highlight other studies that have performed similar studies using the ASTar. However, without reading these other articles in detail the reader cannot identify the similarities/differences between this study and those others, did they all explore the same antimicrobials, did others show similar findings for antimicrobials of interest (i.e. Amoxi-clav; meropenem).
Author response: As far as we are able to tell from what is written in previous papers, all of the antimicrobials tested here were also tested in previous studies. We have already highlighted that the problems with amoxicillin-clavulanate were also reported in a previous study with very similar findings (see lines 248-251). Problems with meropenem have not been previously described and this is likely due to the high numbers of CPE included in simulated samples in our study, as CPE are relatively rare in routine blood cultures. See further details below in our response to a related point.
Polymicrobial cultures were highlighted as the most probable cause for VME’s. Discussion should explore this further by providing more detailed insights into how ASTar differentiates mixed cultures and potential limitations arising from these cases, and/or how this can be addressed in future experiments.
Author response: The problem is that ASTar does not differentiate mixed cultures. However, we agree that further discussion could be of benefit to discuss how the problem might be avoided in routine practice. We have therefore included an additional paragraph of text as follows (lines 240-247):
This problem could potentially be reduced by a more thorough and careful reading of Gram-stained smears to exclude the presence of Gram-positive bacteria. The complete antibiogram should also be fully considered when assessing the validity of results. For example, a polymicrobial blood culture containing E. cloacae complex and E. faecalis showed resistance to meropenem (attributable to E. faecalis) but susceptibility to piperacillin-tazobactam when tested by ASTar. This represents a highly unusual antibiogram for a pure isolate of E. cloacae and should raise suspicions of the possibility of a polymicrobial culture.
While the inclusion of resistant strains is noted, more quantitative data about how ASTar handled complex resistance mechanisms (like β-lactamases) would enhance the robustness. I.e do specific resistance mechanisms more commonly result in ME/VME’s….this is briefly touched on in the results for meropenem and should be discussed in more detail.
Author response: This is an interesting point. Aside from colistin (for which the number of resistant strains was too small to draw conclusions) the only known (characterized) resistance mechanisms were beta lactamases. In Table 6 we can see that there was only one ME or VME for 6 of the beta lactam antibiotics tested - so no useful conclusions can be drawn for these - but for meropenem there were 8 ME and 2 VME and we agree that this warrants further analysis. We have added an additional paragraph to the discussion to address this (see lines 260 - 275) as follows:
From 64 patient's blood cultures, we did not encounter any Gram-negative isolates that were resistant to meropenem and categorical agreement between ASTar and BMD was high (97%). In contrast, when testing 56 simulated blood cultures, there were 8 major errors and 2 very major errors. All of these 10 errors were associated with strains producing a variety of carbapenemases. The two Enterobacterales showing very major errors had ASTar MICs of 4 mg/L (SIE) versus MICs of 16 mg/L (R) when tested by BMD. Major errors were associated with A. baumannii (n = 3), P. aeruginosa (n = 1) and various Enterobacterales (n = 4). ASTar MICs for these isolates were 16-64 mg/L (R) versus MICs of 4-8 mg/L (SIE) when tested by BMD. The reasons for the relatively poor correlation for meropenem when testing carbapenemase-producers are unclear. Esse et al. did not report any specific data for carbapenemase-producers but reported a categorical agreement of 90.7% for meropenem, which was a relatively low when compared to other antibiotics [29]. It is not known whether any carbapenemase-producers were included in the study by Göransson et al. [28] and all isolates in the study by Banchini et al. were susceptible to carbapenems [30]. Further studies with carbapenemase-producing isolates would be of value.
Can the authors posit reasons for the poor performance with Amoxi-clav (ME’s in patient samples), meropenem, gentamicin, ceftazidime, and aztreonam (ME’s in simulated samples), consider ways that these may be overcome, or future studies to address these?
Author response: For co-amoxyclav, we have re-examined the data and it is notable that all 10 of the isolates showing ME were highly resistant to ampicillin (note ampicillin was not one of the agents under study but ASTar MICs for ampicillin were reported by ASTar). These isolates are therefore AMP R but susceptible to co-amoxyclav – inferring beta lactamase production. The clavulanate concentration is therefore critical to neutralise the beta lactamase and restore susceptibility to amoxicillin. As clavulanate is notoriously unstable, we suspect that the concentration of active clavulanate in the ASTar kit is probably lower than intended or required. We are unable to be too definitive about this in the absence of more evidence but we have added additional text as follows (see line 251):
We reported a total of 11 major errors for amoxicillin-clavulanate susceptible isolates when testing patient’s blood cultures using ASTar (E. coli (n = 9), K. pneumoniae (n = 1) and P. mirabilis (n = 1)). Notably, all 11 of these isolates were highly resistant to ampicillin (MIC > 64 mg/L) when tested by ASTar, inferring β-lactamase production, which suggests that the clavulanate component of amoxicillin-clavulanate may be critically relevant to these errors.
For meropenem, we have already commented in some detail with additional text (see earlier responses). We observe a correlation with carbapenemase production but we do not have a reason for the errors. We have identified the need for further work. For “gentamicin, ceftazidime, and aztreonam (ME’s in simulated samples)”, we do not have knowledge of the gentamicin resistance mechanisms to speculate further and for the other agents there is only a single error usually within one dilution of the target MIC.
Reviewer 2 Report
Comments and Suggestions for Authors
In the manuscript, the authors present research on a new system for rapid drug susceptibility testing using the MIC method. For the evaluation, they use cultures from blood infections and cultures from simulated infections. The drug susceptibility results are compared with other methods routinely used in laboratories. The methodology and performance of the experiments raise no objections. The outcomes are correctly interpreted. However, I suggest improving the description of the Results, because without reading the Methods first, it is difficult for the reader to find out about the research carried out.
The results sections are poorly written. The article should lead the reader smoothlythrough the narrative. It's much easier to track results when you add 1-2 introductory sentences.
For example the authors starts with "Table 1 summarizes the 65 Gram-negative species that were
recovered from 64 blood cultures obtained from patients, and their susceptibility to 12 antibiotics is summarized in 81 Table 2" .
It would sound much better if you wrote, for example: "65 Gram-negative species were identified by mass spectrometry from 64
patients blood cultures. The most frequently identified species was Escherichia coli, followed by Klebsiella pneumoniae (Table 1)" e.t.c.
Attention should be paid to the stages and methods used. It is good practice to briefly describe the results and only then refer the reader to the contents of Tables. The article presents and describes the results and summarizes them in tables. In particular, the description of the results of parts 2.1-2.4 should be changed.
Author Response
Responses to reviewer 2
The authors are grateful for your diligent and helpful review. Please see our responses below.
In the manuscript, the authors present research on a new system for rapid drug susceptibility testing using the MIC method. For the evaluation, they use cultures from blood infections and cultures from simulated infections. The drug susceptibility results are compared with other methods routinely used in laboratories. The methodology and performance of the experiments raise no objections. The outcomes are correctly interpreted. However, I suggest improving the description of the Results, because without reading the Methods first, it is difficult for the reader to find out about the research carried out.
The results sections are poorly written. The article should lead the reader smoothly
through the narrative. It's much easier to track results when you add 1-2 introductory sentences.
For example the authors starts with "Table 1 summarizes the 65 Gram-negative species that were
recovered from 64 blood cultures obtained from patients, and their susceptibility to 12 antibiotics is summarized in Table 2" .
It would sound much better if you wrote, for example: "65 Gram-negative species were identified by mass spectrometry from 64 patients blood cultures. The most frequently identified species was Escherichia coli, followed by Klebsiella pneumoniae (Table 1)" e.t.c.
Attention should be paid to the stages and methods used. It is good practice to briefly describe the results and only then refer the reader to the contents of Tables. The article presents and describes the results and summarizes them in tables. In particular, the description of the results of parts 2.1-2.4 should be changed.
Author response: Thank you for your useful comments. We strongly agree that without reading the Methods first, it is difficult for the reader to find out about the research carried out. We would definitely advocate reading the methods first. It is a shame that the formatting requirements of the journal place the results section first. Our main aim was to keep the paper as concise as possible and avoid duplication by having information either in the tables or in the text, but not in both. However, your points are well made and we agree that the paper could be more reader-friendly in accordance with what you have recommended. We have modified the introductions to sections 2.1 – 2.4 as recommended in order to guide the reader through the narrative of the paper as recommended. We believe this improves the readability of the paper. The modified text for each of these sections is shown below:
2.1. Species recovered from patient’s blood cultures and reference MIC results as determined by BMD.
Sixty-five Gram-negative isolates were recovered from 64 patient’s blood cultures. The most frequently identified species were Escherichia coli and Klebsiella pneumoniae. For most of these blood cultures (54/64) an acceptable score for species identification was obtained by performing matrix-assisted laser desorption/ionization-time of flight mass spectrometry (MALDI-TOF MS) directly on blood culture extracts. The remainder were identified by testing colonies from subcultures after 4 h incubation. The species recovered are listed in Table 1 and their susceptibility to 12 antibiotics as determined by measurement of minimum inhibitory concentrations (MICs) using broth microdilution is summarized in Table 2.
2.2. Species recovered from simulated blood cultures and reference MIC results as determined by BMD.
All Gram-negative species that were artificially inoculated into 56 simulated blood cultures were recovered as expected, with no additional unexpected isolates. Forty seven of the isolates were successfully identified by direct testing of blood cultures and the remainder from colonies obtained by 4 h subculture. The dominant species were E. coli and K. pneumoniae. There were equal numbers of aerobic (FA PLUS) bottles and anaerobic (SN) blood culture bottles. Table 3 summarizes the species that were recovered from the 56 simulated blood cultures, and their susceptibility to 12 antibiotics as determined by measurement of MICs using broth microdilution is summarized in Table 4.
2.3. Performance of ASTar for 64 routine positive blood cultures from patients.
We firstly assessed the performance of ASTar for the susceptibility testing of Gram-negative bacteria in 64 patient's blood cultures. Broth microdilution was used as the comparator. Sixty-five Gram negative isolates were recovered with one bottle containing a mixture of E. coli and K. pneumoniae. The performance data are summarized in Table 5.
2.4. Performance of ASTar for 56 simulated blood cultures.
After testing samples from patients, we then assessed the performance of ASTar using 56 simulated blood cultures that had been inoculated with a range of Gram negative species with known mechanisms of resistance to various antibiotics. Broth microdilution was again used as the comparator. The performance data are summarized in Table 6.
Reviewer 3 Report
Comments and Suggestions for Authors
I confirm that I have read the manuscript, but I don't consider myself an expert on the topic. The Methods are correctly described and the results strongly support the conclusions of the research.
Author Response
I confirm that I have read the manuscript, but I don't consider myself an expert on the topic. The Methods are correctly described and the results strongly support the conclusions of the research.
Author response: Thanks you for your review.
Reviewer 4 Report
Comments and Suggestions for Authors
- Please improve introduction, describe clearly why did you study gram negative bacteria exactly in the positive blood samples.
- Bacterial identification methods was described in the materials method section but there is not data related to this experiment or methods in the result section.
- I also suggest to insert statistical data in the tables.
- There is information about the data analysis but I advice to clearly describe statistically describe the obtaining results as well as in tables and in the text.
Author Response
Responses to reviewer 4
The authors are grateful for your helpful review. Please see our responses below.
- Please improve introduction, describe clearly why did you study gram negative bacteria exactly in the positive blood samples.
Author response: The aim of the study was to evaluate the ASTar instrument. The ASTar instrument is only capable of testing Gram-negative bacteria. We have attempted to emphasise the importance of Gram-negative bacteria in this setting. For example, in line 30 we state that “Gram-negative bacteria are frequently implicated, for example, in a study of 14,000 patients in intensive care in 75 countries, Gram-negative bacteria were isolated from 62% of patients with severe sepsis who had positive blood cultures”.
2. Bacterial identification methods was described in the materials method section but there is not data related to this experiment or methods in the result section.
Author response: Bacterial species were identified by MALDI-TOF MS, which is very well established in UK diagnostic laboratories as the standard method for species identification. The extraction method for preparation of blood cultures can vary between laboratories and we have therefore described this in detail in the methods section. We do not believe it should be necessary to describe the intricacies of the MALDI-TOF MS process as these are very well known as part of a standard routine method.
We have added information to the results section (lines 80-84 and 105-107) to report how many isolates were successfully identified using the direct extraction method and how many isolates required identification from colonies obtained by subculture.
3. I also suggest to insert statistical data in the tables.
Author response: For the majority of scientific papers we agree that it is accepted practice that numerical data should be analysed by an appropriate statistical method. However, the methods for data analysis when comparing different methods of AST are clearly defined by the US FDA with standard formulae for calculation of EA, CA, ME, VME, etc…. This is the approach used by the majority of published studies that compare different methods of AST. We are not convinced that application of any additional statistical tool would provide any further useful insights. However, we remain open to any specific suggestions.
4. There is information about the data analysis but I advice to clearly describe statistically describe the obtaining results as well as in tables and in the text.
Author response: Please see our response to point 3 (above).
Round 2
Reviewer 1 Report
Comments and Suggestions for Authors
Thank you to the authors for their quick response, revisions, and for addressing the comments provided. I appreciate the effort made to expand on the discussion, which has strengthened the manuscript. I have no further concerns, and I believe the revisions have improved the clarity and impact of the work.